# Towards Improving Households' Investment Choices in Tanzania: Does Financial Literacy Really Matter?

**Josephat Lotto**

Department of Accounting and Finance, The Institute of Finance Management, Dar es Salaam 255, Tanzania; josephat.lotto@ifm.ac.tz

**Abstract:** This paper primarily aims to assess the impact of financial literacy on households' investment choices. The paper employs secondary data from the FinScope survey (2017) conducted by Financial Sector Deepening Trust (FSDT). In particular, the study aims at establishing whether the choices of investment platforms are influenced by the financial literacy level of the heads of households. To do so, the study employed both bivariate and multivariate analytical techniques. The study finds that financial literacy is positively and significantly associated with household investment choices. More specifically, as households become more financially literate, they divert from investing in informal groups towards more formal investment platforms such as investment accounts, agricultural ventures as well as personal business. Such observations may be partly attributable to the facts that individuals whose financial literacy is sound enough are more likely to be equipped with skills and knowledge of risks associated with investment opportunities and some other several financial products. The study also reveals that financial literacy is significantly associated with households' socio-demographic factors, and that the adult population exhibits a large financial literacy gap and, therefore, adults should not be considered as a homogenous group—instead, gender, age, education and income levels of the households, which are showcased in this study, should also be taken into consideration. The study opines that, because most of households, as revealed in the survey from which the employed dataset is based, are hailing from rural settings where agriculture is the main economic activity, we establish that agricultural ventures require a complete revamp for Tanzania to become a middle-income economy through its industrialization agenda. The study also proposes the financial literacy programmes to be rolled on to students from early stage of their education such as secondary schools.

**Keywords:** age; gender; education; financial literacy; investment choices

## 1. Introduction

The world economy is currently going through some economic challenges such that every individual needsto be active and smart in investment decisions so as to cater for the rising cost of living. Many individuals consider investments to be captivating because they make decisions and later see the outcomes of the decisions they make (Awais et al. 2016). So ideally, everybody contributes toone form of investment or another; even those who do not participate in investment activities related to buying and selling of financial instruments and other related assets still take on investments through participation in other forms, for instance, pension plans and employee savings programs, buying life insurance, real estate investments and investment in bank fixed deposits (Gallery et al. 2011). However, investments in any form require a sufficient level of financial education, which is a challenge to most individuals who would wish to engage in investment ventures. The role of financial education in improving decision making abilities of individuals has long been recognized across the world. In actual fact, financial literacy helps households in making sound and informed investment decisions,



and this may lead to future income and subsequently to economic growth, as clearly determined by Claessens et al. (2009). According to Claessens et al. (2009) financially literate households, as opposed to their counterparts, illiterate ones, have a greater ability to access financial services, which may enable them to improve their economy byinvesting in education and health (contributing to human capital), starting business ventures and expanding existing investments.

Generally, sophisticated investment decisions such as on the type of financial assets to invest in, and the type of financial institutions to engage with all require a reasonablelevel of financial literacy for one to make viable decisions (Lusardi 2008). Studiessuch as Hastings and Mitchell (2011) emphasize the position of financial literacy in the economy. According to these scholars, financial literacy contributes very much topoverty eradication, improving the living standards of households, improving the market efficiency and ultimately improving economic development.

The United Nations' Sustainable Development Goals (SDGs) purposely recognized the importance of financial literacy by pronouncing goal number four to be"ensuring inclusive and equitable quality education and promote life-long learning opportunities for all." The purpose of this goal is to provide direction to countries towards improved anfinancial literacy level of users and providers of financial services by inspiring these countries to integrate financial education into their local curricula by the end of 2020.

Tanzania hasnot been left behind in taking the direction shown by the United Nations' Sustainable Development Goals (SDGs). It is categorically highlighted in the Tanzania Development Vision (2025) that financial illiteracy hampers access to financial services, which negatively affects the country's economic competitiveness in the global market. To implement this vision, through the Bank of Tanzania, Tanzania established the national financial education framework in 2016 with the sole aim of boosting levels of financial literacy among households (Bank of Tanzania 2016).

Apparently, in Tanzania there is a rapidly growing trend of savings and lending groups among households, especially women. Most of the group members in such savings and lending groups usually end up taking-over members' assets due to their failure to repay the borrowed money from the group. One of the pressing reasons for failure to repay such debts is associated with the wrong choice of investment avenues by members after borrowing the money, due to a lack of proper financial literacy.

Some of the research on financial literacy is not consistent as to whether the financial education programs improve financial decision-making competence of the beneficiaries (Al-Tamimi and Kalli 2009; Gallery et al. 2011; Brown and Graf 2013). Of more interest is the topic of financial literacy in Tanzania where, regardless of the inauguration of National Financial Education Framework the majority of Tanzanians are still financially illiterate and relatively poor, and there is a scarcity of scholarly works which investigate the value of financial literacy in improving mobilization of wealth among Tanzanian households through investment ventures. This study aims at examining the influence of financial literacy on households' investment choices. We use the credible dataset from FinScope (2017) with their respective definitions of financial literacy and investment choices. To the best of the author's knowledge, this study is among the first of its kind to be conducted in Tanzania. The findings of the paper have a significant implication for financial education and public policy programs.

## 2. Literature Review

### 2.1. Theoretical Underpinnings

OECD (2005) defines financial literacy as "the process by which financial consumers/investors improve their understanding of financial products, concepts and risks, and through information, instruction and/or objective advice, develop the skills and confidence to become more aware of the financial risks and opportunities to make informed choices to know where to go for help and to take other effective actions to improve their financial wellbeing". Thus, it can be correctly stated that financial literacy improves households' ability to know, monitor and effectively use financial resources to enhance their economic well-being. The measurement of financial literacy is not an easy task.

According to Lusardi and Mitchell (2011), as quoted "While it is important to assess how financially literate people are, in practice it is difficult to explore how people process economic information and make informed decisions about household finances". Other definitions are also available in the literature. According to Beverly et al. (2003), financial literacy is the ability of an individual to possess financial knowledge. Mandell (2008) considers financial literacy as the ability to evaluate the new and complex financial instruments and make informed judgments on both the choice of instruments and extent of use that would be in their own best long-run interests. Also, we have Schagen (2007) who defines financial literacy as the ability to make informed judgements and to take effective decisions regarding the use and management of money. Furthermore, Nguyen and Rozsa (2019) categorize the definition of financial literacy into basic literacy, which includes the individual's understanding of basic financial concepts such as compound interest, inflation, the time value of money, the money illusion and the advanced financial literacy, which calls for understanding of individuals of financial issues such as risky assets, long-term period returns, volatility, diversification, asset allocation, performance, risk rating and return rating.

Investment choice is one of the important decisions when it comes to investment ventures done by households across the world. Consequently, empirical studies confirm that financial education is paramount in investment choice decisions (Falk et al. 2010; Rooij et al. 2011; Brown and Graf 2013). Literature recognizes the importance of financial literacy in investment choices, and several studies (Al-Tamimi and Kalli 2009; Brown and Graf 2013) have established a relationship between financial literacy and investment choices in different markets ranging from financial markets to physical markets, which ultimately results in an improvement in the living standards of households and thereby leads to economic growth in developing countries, Ellis et al. (2010). Financial illiteracy among households results in an inability to manage savings and investments.

On the one hand, financial literacy aids households in making well-informed financial decisions concerning investments and savings, and also makes them able to mitigate prospective financial risks linked with financial services and products. On the other hand, financially literate households ordinarily demand more financial services and products, consequently leading to reinforcement of the financial market firmness or stability and remarkably leading into economic growth and development (Ellis et al. 2010).

## 2.2. Empirical Literature

It is indisputable that a low level of financial literacy leads individuals to make sub-optimal economic choices and commit financial mistakes. On the asset side of the household balance sheet, poor financial literacy affects saving and investment decisions, accumulation of wealth, access to financial markets and portfolio choices. In particular, a poor level of financial literacy is related to lower savings and wealth accumulation before retirement (Lusardi and Mitchell 2007; Clark and Strauss 2012; Bernheim and Garrett 2012), leads investors to choose high-fee investment funds (Hastings and Tejeda-Ashton 2008; Hastings and Mitchell 2011), reduces access to financial markets and stockholdings (Christelis et al. 2010; Rooij et al. 2011; Klapper and Lusardi 2013; Cole and Andrew 2014) and induces sub-optimal portfolio diversification (Guiso and Jappelli 2009; Abreu and Mendes 2010; Santos and Abreu 2013). On the other hand, onthe liability side, poor financial literacy influences financing decisions in terms of funding costs, refinancing choices and risk of over-indebtedness and financial distress. Lower levels of financial literacy are associated with higher mortgage fees (Campbell 2006; the use of more expensive financing alternatives Agarwal et al. 2009), over-indebtedness (Stango and Zinman 2009; Lusardi and Tufano 2008) and mortgage delinquency. The social negative outcome of poor financial literacy is, therefore, financial fragility (Lusardi and Mitchell 2007; Lusardi and Tufano 2008).

Some research work, for instance Christelis et al. (2010), insist on the importance of financial literacy in preventing households from making poor financial decisions. According to the authors, a higher level of financial literacy increases level of households' living standard. Furthermore,

poor financial literacy leads to negative credit behavior, which causes higher indebtedness and debt accumulation problems, high-cost borrowings and loans and making unideal choices form ortgages and other financial products (Lusardi and Tufano 2008).According to Lusardi and Tufano (2008), one may claim that financially literate consumers make better financial decisions, have more chances to invest on stock markets, diversify risk, obtain cheaper borrowing and mortgages, avoid getting into debts, manage efficiently their investments, plan for retirement and accumulate more retirement savings.

Previous empirical studies have established a strong relationship between financial literacy and different perspectives of investment decisions (Al-Tamimi and Kalli 2009; Gallery et al. 2011; Brown and Graf 2013). According to literature (Al-Tamimi and Kalli 2009; Gallery et al. 2011; Brown and Graf 2013), successful financial decision making is the function of different factors including financial information, residence of the households and demographic factors of the households. The investment behavior of the investors can be influenced by the level of financial information that the decision maker possesses. Chong and Lai (2011) describe financial information to include accounting reports, general information about price movements, a firm's reputation, status of a firm in the investment market, past performance of the firm's stock as well as the expected performance of the firm. According to Chong and Lai (2011), when implementing a particular investment decision, decision makers seek information on a firm's performance as well as the investment behavior of other investors in the market, and that the timing and delivery of such market information is very crucial in influencing how investors make their decisions.

Similarly, findings of previous studies from developed economies such as Lusardi and Mitchell (2006) and Rooij et al. (2011) reveal that informal sources of financial information e.g., information from friends and families do not necessarily make individuals financially literate. Likewise, those individuals who have high level of financial literacy mostly rely on formal financial advice like that provided by professional financial advisors.

## 3. Methodology

### 3.1. Data

This paper employs secondary data from the FinScope survey (FinScope 2017) conducted by Financial Sector Deepening Trust (FSDT) in collaboration with the Bank of Tanzania (BOT), National Bureau of Statistics (NBS) and Ministry of Finance and Planning (MoF). This is a national survey representative of adult individuals living in Tanzania. The survey considers an adult to be any Tanzanian who is 16 years or older at the time of conducting the survey. The survey targeted 1000 enumeration areas (EA) from five regions in the Tanzania mainland, namely Iringa, Singida, Mtwara, Rukwa and Mwanza. However, only 998 enumeration areas were reached to interview 9459 respondents from the sample of 10,000 respondents. In addition, because the focus of this study is on the household level, the analysis data was collapsed to 3812 households, limiting respondents to the heads of the households.

### 3.2. Variable Description

Financial Literacy

In this paper, financial literacy is measured in terms of an elementary financial understanding of three key components: interest rates, discounting and borrowing, as used in a FinScope (2017) survey where the data of this study are adapted from. This approach is also proposed by Lusardi and Mitchell (2011), where three questions are asked to the households to test their knowledge of the aforementioned concepts, including interest rates, discounting and borrowing. Each correctly answered question is awarded one point, while an aggregate score is calculated by taking the average of the results of the three questions, and is considered as the measure of the financial literacy, termed AGG_FL (Aggregate Financial Literacy).AGG_FL is an ordinal variable which takes the value of 1 if the respondent answers

all three questions correctly and therefore is considered financially literate, and is 0 otherwise. The rest of the variables are described in Table 1 below.

**Table 1.** Variables Description.

| S/n | Variable | Description | Nature |
|-----|----------|-------------|--------|
| 1. | Investment Choices | i.   Informal Groups<br>ii.  Agriculture<br>iii. Personal Business<br>iv. Investment Account | *Categorical variables*;<br>In each investment choice, the variable takes the value 1 if the choice is either Informal, Agriculture, Personal Business or Investment Account; Otherwise, the variable takes the value 0 for each respective choice. |
| 2. | Age (*age*) | Number of years lived by the household head | *Continuous Variable* |
| 3. | Gender (*gender*) | Sex of the head of the household | *Dummy variable:* It takes the value of 1 1 if the head of the household is male and 0 if female. |
| 4. | Income (*hhincome*) | Represents household annual income | *Continuous Variable* |
| 5. | Education level (*edu*) | Represents highest level of education reached by the head of the household.<br>i.   No Formal Education<br>ii.  Primary<br>iii. Secondary<br>iv. Tertiary | *Categorical variables:* 1 if no formal education, 0 otherwise; 1 if primary education, 0 otherwise; 1 if secondary education, 0 otherwise;<br>1 if tertiary education, 0 otherwise. |
| 6. | Employment (*emp*) | Represents an employment status of head of the household whether he/she is employed or not | It is a dummy variable with a value of 1 if the head of the household is employed and 0 otherwise. |
| 7. | Information (*info*) | Indicates the source of financial-related information obtained by head of the household whether formal or informal | This is a dummy variable with the value of 1 if the source of information regarding financial matters is formal and 0 for informal sources |
| 8. | Location (*loc*) | This denotes the head of the household's place of residence | It is a dummy variable coded as 1 if urban and 0 if rural. |

*3.3. Analytical Design*

A binary probit regression model was employed in this study where household investment choices (dependent variable) are considered as discrete choices. The assumption used in this model is that the error term is normally distributed with a mean of zero and a unitary standard deviation as articulated by Greene (2012). The probit model to examine the effects of financial literacy as well as other independent variables on the household investment choice was specified as follows:

$$pr(choice_i = 1) = \varnothing(\beta_0 + \beta_1 AGG\_FL_i + \beta_2 hhincome_i + \beta_3 loc_i + \beta_4 edu_i + \beta_5 emp_i + \beta_6 info_i + \beta_7 gender_i + \varepsilon_i)$$

where

$choice_1$ = Informal groups
$choice_2$ = Investment account
$choice_3$ = Household personal business

$choice_4$ = Agricultural investment

$\varepsilon_i$ is the error term

$Info_i$ is the information

$gender_i$ is the gender of the household head $i$

$edu_i$ is the highest education level attained by the household head $i$

$loc_i$ is the location or household head's place of residence.

$emp_i$ is the employment status of the household head $i$

$hhincome_i$ denotes household head $i$'s income

$\varepsilon_i$ is the error term

## 4. Results and Discussions

*Multivariate Analysis*

This paper employed probit regression to assess whether financial literacy influences households' investment choices in Tanzania, while controlling for households' demographic and socioeconomic factors. Various diagnostic tests were conducted such as a multicollinearity test and heteroscedasticity test. In order to test for the presence of multicollinearity, Variance Inflation Factors (VIF) and Pearson correlation analysis were employed. The analysis found a mean VIF of 3.06, which is far below the cut-off point of 10 as suggested by Belsley et al. (1980) which determines whether there is serious multicollinearity. According to the cut-off point, the VIF reported that multicollinearity is not a problem. After testing for multicollinearity, a Breusch-Pagan test for heteroscedasticity was conducted. The fear in testing for heteroscedasticity is the existence of homogeneity of variance of the residuals. This is one of the conditions to be observed before employing and multivariate regression analysis. The results of Breusch-Pagan test show a Chi square value below the critical value, implying that the hypothesis for homoscedasticity could be accepted. Likewise, the heteroskedasticity test shows that the variances of the OLS estimators are not biased.

In order to assess the link between financial literacy and household investment choices and decisions, probit regression was run with investment choice as a dichotomous dependent variable and financial literacy as an independent variable. This was followed by socio-demographic characteristics as control variables. The results in Table 2 show that financial literacy has a strong and positively statistically significant relationship with the investment choices. The relationship is statistically significant at the 1% significance level. The study further shows that 82% of financially literate head of households are more likely to invest in investment accounts, while 47% of them would more likely prefer personal business accounts. The study also shows that 42% of the literate heads of households are more likely to invest their proceeds in agricultural projects, while about 43% of these household heads would choose to invest their money in informal groups. These results imply that as households become more financially literate, they divert from investment in informal groups towards formal investment accounts, agricultural investment as well as personal business. This observation can partly be attributed to the fact that financial literacy increases understanding of risks associated with investment ventures and knowledge on financial products, as well as other profitable investment ventures. The findings are consistent with Rooij et al. (2011) who found out that financially literacy among households increases participation in the stock market.

The results further reveal that urban households are about 13% and 9% more likely to invest in investment accounts and personal business, respectively, as compared to their counterparts, rural households. Likewise, urban households are approximately 7% and 3% less likely to invest in informal groups and agricultural investment, respectively, as opposed to their counterparts, rural households. The explanation of this finding can partly be supported by the reality that urban households may easily access financial products due to the presence of a greater number of financial institutions located in urban compared to rural areas, and that urban households rarely participate in agricultural activities due to lack of enough land in townships. The results also show that households in urban areas are

more likely to invest in personal businesses because it is believed that demand for business goods as well as services is usually higher in urban areas than in rural areas. This finding is in agreement with Cole et al. (2009).

**Table 2.** Probit Regression Results.

| VARIABLES | Informal Groups | Investment Account | Household Personal Business | Agricultural Investment |
|---|---|---|---|---|
| Financial literacy | −0.431 ** | 0.817 *** | 0.468 *** | 0.418 *** |
| | (0.202) | (0.130) | (0.101) | (0.112) |
| Location | −0.0711 *** | 0.133 *** | 0.0935 *** | −0.260 *** |
| | (0.0231) | (0.0154) | (0.0119) | (0.00996) |
| Gender | −0.0301 | −0.00656 | −0.0253 ** | 0.0382 *** |
| | (0.0238) | (0.0153) | (0.0116) | (0.0130) |
| Primary | −0.0764 *** | 0.155 *** | 0.0335 ** | −0.0922 *** |
| | (0.0286) | (0.0151) | (0.0138) | (0.0161) |
| Secondary | −0.236 *** | 0.280 *** | 0.0162 | −0.226 *** |
| | (0.0418) | (0.0288) | (0.0221) | (0.0221) |
| Tertiary | −0.320 *** | 0.617 *** | 0.0520 | −0.129 *** |
| | (0.0568) | (0.112) | (0.0344) | (0.0307) |
| Employment status | −0.161 *** | 0.227 *** | −0.264 *** | −0.0854 *** |
| | (0.0355) | (0.0375) | (0.0340) | (0.0194) |
| Information | −0.177 * | 0.122 | −0.0330 | −0.0180 |
| | (0.0986) | (0.0819) | (0.0576) | (0.0570) |
| Observations | 3812 | 3812 | 3812 | 3812 |
| Wald Chi2 (9) | 195.03 | 525.32 | 178.70 | 915.03 |
| Significance | 0.0000 | 0.0000 | 0.0000 | 0.0000 |
| Pseudo R2 | 0.0831 | 0.1462 | 0.0724 | 0.2483 |
| % correctly classified | 63.65 | 71.65 | 86.31 | 79.17 |

Note: Robust standard errors in parenthesis. *** $p < 0.01$, ** $p < 0.05$, * $p < 0.1$.

The gender of household heads is an important parameter that influences the investment choices at the household level. The results in Table 2 show that male household heads are about 3% less likely to invest in personal businesses, but are about 4% more likely to invest in agricultural investments relative to female household heads. These findings indicate that men are more risk averse and less likely to invest in the informal groups, investment accounts as well as personal businesses. Based on the same argument, men are more likely to participate in agricultural activities relative to women.

Regarding the education level of heads of households, results reveal that heads of households with only primary education are approximately 8% and 9% less likely to invest in informal groups and agricultural investment, respectively, but are 16% and 3% more likely to invest in investment account and personal business, respectively, compared to households with no formal education. Similarly, household heads with a secondary education are approximately 24% and 23% less likely to invest in informal groups and agricultural investments, respectively, but are 28% more likely to invest in investment accounts compared to households with no formal education. Also, household heads with a tertiary education are approximately 32% and 13% less likely to invest in informal groups and agricultural investment, respectively, but are 62% more likely to invest in investment accounts and personal businesses, respectively, compared to households with no formal education. The message which one can derive from these findings is that an increase in education level lowers the likelihood of

investing in informal groups as well as agricultural ventures. These findings are in line with Rooij et al. (2011).

The results, further, reveal that employed household heads are approximately 16%, 26% and 9% less likely to invest in informal groups, personal businesses and agricultural investment, respectively, but are 23% more likely to invest in investment accounts relative to unemployed household heads. These results are consistent with Calderone (2014). This study further reveals that household heads that access financial advice from formal sources are approximately 18% less likely to invest in informal groups relative to household heads with access to informal sources of financial information. This can partly be attributed to the fact that formal financial advice increases financial literacy among households.

## 5. Conclusions

This paper primarily aims at assessing the impact of financial literacy on enhancing households' investment choices. The results of the paper reveal that as households become more financially literate, they divert from investing in informal groups towards more formal investment platforms such as investment accounts, agricultural investment as well as personal businesses. Such observations may be partly attributed to the facts that individuals whose financial literacy is sound enough are in a way equipped with skills and knowledge of risks associated with investment opportunities and other several financial products.

The study also reveals that financial literacy is significantly associated with household socio-demographic factors, and that the adult population exhibits a large financial literacy gap; therefore, adults should not be considered as a homogenous group. Instead, gender, age, education and income levels of households, which are showcased in this study, should also be taken into consideration. The study indicates that because most of households, as reflected in the survey from which the adapted dataset is based, are hailing from rural settings, and because in their setting agriculture is the main economic activity from which the source of their income is derived, we establish that agricultural ventures require a complete revamp for Tanzania to become a middle-income economy through its industrialization agenda. The paper also proposes for financial literacy to be a mandatory program for the secondary schools curriculum in Tanzania as practiced elsewhere, for example in the Czech Republic and Slovakia, because appropriate management of personal finance creates preconditions for a successful and high quality life.

**Funding:** This research received no external funding.

**Conflicts of Interest:** The Author declares that there is no conflict of interest.

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
