# Peer review of "Towards Improving Households’ Investment Choices in Tanzania: Does Financial Literacy Really Matter?"

_ijfs, doi:10.3390/ijfs8020029_

Round 1
Reviewer 1 Report
Dear authors,
there are some universal issues associated with the writing of a scientific paper. The paper should be divided as follows: Introduction, Literature review, (Materials)and Methods, Results and Discussions, Conclusion (see the section Instruction for authors). Please, use IJFS template for your article!
https://www.mdpi.com/journal/ijfs/instructions
Focus on Discussion part! This section should compare the results achieved with the other ones to highlight the importance of the discussed topic.
In my opinion, it is possible to publish your paper after correcting issues mentioned above.
I hope, my comments may help you to improve the paper.
Author Response
REVIEWER ONE
COMMENT 1:
There are some universal issues associated with the writing of a scientific paper. The paper should be divided as follows: Introduction, Literature review, (Materials) and Methods, Results and Discussions, Conclusion
RESPONSE 1:
The attached revised version of the paper is now structured as recommended by the reviewer
COMMENT 2:
Focus on Discussion part! This section should compare the results achieved with the other ones to highlight the importance of the discussed topic.
RESPONSE 2:
The section is revised to accommodate the reviewer’s comment. The theoretical implication/importance of the discussed issues are highlighted in red while the literatures which support the findings are highlighted in blue
Reviewer 2 Report
Referee report for Towards Improving Households’ Investment Choices in Tanzania: Does Financial Literacy Really Matter?
Summary: According to the abstract, the authors examine the socioeconomic determinants of financial literacy and how that impacts financial choices. They use Finscope Survey of 2017 as their source of data. They find that there is heterogeneity with respect to financial literacy. As financial literacy increases households switch from informal to formal market investments.
Overall comments
- Your findings are important and contribute to the literature. The connection between financial literacy and financial behaviors is a growing field and consensus has not been developed. Your sample sheds light on a population that data is hard to obtain.
- The biggest limitation to this research is the writing. It is hard to follow the contribution because of extra words or run on sentences. I recommend a thorough editing of the paper to help make it concise and to the point.
- Your introduction reads more like a lit- review rather than an introducing the problem and your findings.
- Multi collinearity test, heteroskedasticity, and model specification can be taken out. We assume that you ran the test, and you can include a sentence or two about you running the tests and finding support for your methodology.
- Results-
- The residuals from the first model as saved and used to generate RFL? Why? I am not sure I am clear about why this is necessary. I don’t see future reference to it.
Suggestions
Even as mid career economist, I often revisit this paper on writing tips. https://faculty.chicagobooth.edu/john.cochrane/research/papers/phd_paper_writing.pdf
I am sharing it with you hoping you find it as helpful as I do for my own work.
Author Response
REVIEWER TWO
COMMENT 1
Your findings are important and contribute to the literature. The connection between financial literacy and financial behaviors is a growing field and consensus has not been developed. Your sample sheds light on a population that data is hard to obtain.
RESPONSE 1
I appreciate the complement from the reviewer
COMMENT 2
The biggest limitation to this research is the writing. It is hard to follow the contribution because of extra words or run on sentences. I recommend a thorough editing of the paper to help make it concise and to the point.
RESPONSE 2
The current version of the paper is revised and edited to reduce the volume, and it is now concise and to the point.
COMMENT 3
Your introduction reads more like a lit- review rather than an introducing the problem and your findings.
RESPONSE 3
The introduction is now edited carefully to only introduce the research problem, and all information which seems to be of literature review nature are taken to the literature review section on page 4 and highlighted in purple color in an attached reviewed version of the paper
COMMENT 4
Multi co linearity test, heteroskedasticity, and model specification can be taken out. We assume that you ran the test, and you can include a sentence or two about you running the tests and finding support for your methodology.
RESPONSE 4
The recommendation of the reviewer is observed. Multicolinearity &heteroskedasticity tests and model specification are taken out, and I have included only key findings of the tests. This is highlighted in green and found on page 9 of the attached reviewed version of the paper
COMMENT 5
The residuals from the first model as saved and used to generate RFL? Why? I am not sure I am clear about why this is necessary. I don’t see future reference to it.
RESPONSE 5
I have reviewed the paper to specify only the model of Investment Choices in relation to financial literacy as an independent variable along side with the control variables (socio-demographic characteristics of the households). The issue of saving the residuals from the first model and using it to generate RFL does not exist anymore. This revision run from page 8 to page 9 of the attached reviewed version of the paper
Suggestions
I have looked at the guidelines for paper writing provided by the reviewer and found it very useful to shape the revised version of my paper. This guideline is found in the link below;
https://faculty.chicagobooth.edu/john.cochrane/research/papers/phd_paper_writing.pdf
Reviewer 3 Report
Comments
The author undertook an analysis of the significant topic. Financial literacy is important in many aspects of the economy. Choosing the issue and showing the importance of financial literacy in the economy, in the sustainable development and development of financial services is most appropriate and desirable.
An Introduction on p. 3 The author writes: "Literature recognizes the importance of financial literacy in investment choices, and several studies (...)". It is reasonable to present the sources to which the author refers. The general statement that some sources write so is insufficient.
Although most of Journal's focus is on the global approach to the phenomenon being studied, I think that looking from one country's scale also has added value because it allows you to see more details that cannot be seen from a macro point of view, from a global scale, which is why Research results only from Tanzania can have added value.
Analysis of the material, however, indicates some gaps in understanding the concept of financial literacy. According to the literature on the subject, financial literacy includes in its scope: financial knowledge, financial skills, financial behavior, financial attitude and in some items financial information demand.
If the author has a different definition approach, one should first present how the definition of financial literacy is presented in the literature on the subject, and then present the author's definition proposal together with argumentation and justification.
I believe that the article is methodically underdeveloped:
- on the one hand, a misunderstanding of the concept of financial literacy and a lack of specification of this concept in the implementation;
- reading the article leads to the conclusion that the author identifies financial literacy with financial knowledge, which is a significant narrowing of the phenomenon studied;
- there is no visible research problem reinforced by a clearly defined goal, presenting hypotheses that the Author intends to prove and research questions to which he intends to find answers.
On page 4, the author writes: "Furthermore, poor knowledge on financial education leads to negative credit behavior (...). The sentence is incomprehensible. What does the author mean by" poor knowledge on financial education "? in finance, not financial education.
In the literature review, the author widely illustrates the impact of financial literacy household finance, including retirement planning, in general on financial decisions. The article concerns investment decisions and the author's goal should be to review literature in terms of financial literacy and investment decision, investment choice, and not as wide as it is in its current state. Therefore, the text requires refinement in this area and adaptation of the literature review to the title proposed by the author.
p. 5 The author writes: "According to literature (...)." There is no entry in parentheses, according to which authors, when?).
Footnotes included in the text are incomplete. There is a lack of literature in the list of cited in footnotes: Falk et al, 2010; Bailey et al, (2003); Al-Tamimim & Bin Kalli, (2009); Worthington, (2008); Alessie et al, (2011); Dvorak & Hanley (2010). No reference list. Lusardi & Mitchell, (2007) in the reference list is as Lusardi & Mitchell, (2007a).
Literature cited in the text and included in the bibliography (Brown, M., and Graf, R., (2012). Financial Literacy, Household Investment and Household Debt: Evidence from Switzerland. Swiss Institute of Banking and Finance (S / BF -HSG) Working papers on Finance No. 13/1.) is from January 2013, not 2012.
In 3.2 Variable Description Author choose the method used by Lusardi and Mitchell (2011a). The choice of three questions is beyond doubt. However, the choice of questions raises reservations. The author decided to apply questions regarding interest rates, discounting and borrowing. The article deals with investment choices of households, why did the author choose these questions and not others? Also, in the theoretical part, the author devotes a lot of space to the risk of an investment that is not addressed here. There is a visible lack of consistency between the title of the article and questions, as well as a description in the theoretical part and the empirical part.
I consider the choice of the Binary probit regression method reasonable. The analysis is impressive and is made at a better level than the theoretical part. However, reading the article indicates some inaccuracies between the theoretical and empirical parts. The theoretical part is quite general. It tackles investment issues, but not as much as the theoretical part.- The article requires significant refinement and improvement. At the moment, it can be treated more like a research report than a scientific article.
Author Response
COMMENTS FOR REVIEWER 3
COMMENT 1
The author undertook an analysis of the significant topic. Financial literacy is important in many aspects of the economy. Choosing the issue and showing the importance of financial literacy in the economy, in the sustainable development and development of financial services is most appropriate and desirable.
RESPONSE 1
I appreciate the complement from the reviewer
COMMENT 2
An Introduction on p. 3 The author writes: "Literature recognizes the importance of financial literacy in investment choices, and several studies (...)". It is reasonable to present the sources to which the author refers. The general statement that some sources write so is insufficient.
RESPONSE 2
This phrase is now in literature review page 4 of the revised version. The recommended sources by the reviewer are provided.
COMMENT 3
Although most of Journal's focus is on the global approach to the phenomenon being studied, I think that looking from one country's scale also has added value because it allows you to see more details that cannot be seen from a macro point of view, from a global scale, which is why Research results only from Tanzania can have added value.
RESPONSE 3
I appreciate the complement from the reviewer
COMMENT 4
Analysis of the material, however, indicates some gaps in understanding the concept of financial literacy. According to the literature on the subject, financial literacy includes in its scope: financial knowledge, financial skills, financial behavior, and financial attitude and in some items financial information demand.
RESPONSE 4
The gap in understanding the concept of financial literacy identified by the reviewer does not exist anymore as various definitions of the financial literacy are provided in the attached revised version of the paper found on page 4 of the theoretical literature part and highlighted in blue color
COMMENT 5
If the author has a different definition approach, one should first present how the definition of financial literacy is presented in the literature on the subject, and then present the author's definition proposal together with argumentation and justification
RESPONSE 5
Different definitions are provided on page 4 of the revised version of the paper and the proposed definition used in the study is justified on page 7 and highlighted in yellow color
COMMENT 6
I believe that the article is methodically underdeveloped:
- On the one hand, a misunderstanding of the concept of financial literacy and a lack of specification of this concept in the implementation;
- reading the article leads to the conclusion that the author identifies financial literacy with financial knowledge, which is a significant narrowing of the phenomenon studied;
- There is no visible research problem reinforced by a clearly defined goal, presenting hypotheses that the Author intends to prove and research questions to which he intends to find answers.
RESPONSE 6
- The concept of financial literacy is now well elaborated and different aspects of the term are well covered in the theoretical literature part of the paper and found on page 4 of the revised version of the paper.
- Financial literacy definition used is borrowed from the FinScope (2017) survey where the full set of data used in the study is borrowed from there, so it could not be easy expand the scope of the definition. I believe the use of this definition does not compromise the results presented and conclusion drawn from this paper. In future paper a broader definition will be considered.
- The research problem and the objective of the paper are now clearly shown on the introduction part of the paper and presented in page 2 of the attached revised paper and highlighted in green color
COMMENT 7
On page 4, the author writes: "Furthermore, poor knowledge on financial education leads to negative credit behavior (...). The sentence is incomprehensible. What does the author mean by" poor knowledge on financial education "? in finance, not financial education.
RESPONSE 7
The observation of the reviewer is appreciated and the comment incorporated in page 5 of the attached revised version of the paper and highlighted in light blue color
COMMENT 8
In the literature review, the author widely illustrates the impact of financial literacy household finance, including retirement planning, in general on financial decisions. The article concerns investment decisions and the author's goal should be to review literature in terms of financial literacy and investment decision, investment choice and not as wide as it is in its current state. Therefore, the text requires refinement in this area and adaptation of the literature review to the title proposed by the author.
RESPONSE 8
The comment of the reviewer is very relevant. I have improved my literature by sticking on only the relevant literature, and that irrelevant literature such as on retirement planning is removed as may be reviewed from the attached revised version of the paper in page 5
COMMENT 9
- 5 The author writes: "According to literature (...)." There is no entry in parentheses, according to which authors, when?).
RESPONSE 9
The literature required is quoted in page 6 of the revised version of the paper and written in green
COMMENT 10
Footnotes included in the text are incomplete. There is a lack of literature in the list of cited in footnotes: Falk et al, 2010; Bailey et al, (2003); Al-Tamimim & Bin Kalli, (2009); Alessie et al, (2011); Dvorak & Hanley (2010). No reference list. Lusardi & Mitchell, (2007) in the reference list is as Lusardi & Mitchell, (2007a).
RESPONSE10
This referencing problem is solved.
- The references which were missing in the reference list are now included and highlighted in yellow (Falk et al, 2010; Bailey et al, (2003)) found in page..... and page ....... of the attached revised version of the paper
- Some of them (Alessie et al, (2011); Dvorak & Hanley (2010, Worthington, (2008); Al-Tamimim & Bin Kalli, (2009);) are deleted as their corresponding literatures no longer fit the paper in its revised version form as attached
- Lusardi & Mitchell, (2007) now is similar in both text and the reference list
COMMENT 11
Literature cited in the text and included in the bibliography (Brown, M., and Graf, R., (2012). Financial Literacy, Household Investment and Household Debt: Evidence from Switzerland. Swiss Institute of Banking and Finance (S / BF -HSG) Working papers on Finance No. 13/1.) Is from January 2013, not 2012.
RESPONSE 11
This amendment is done and highlighted in yellow and found in page 13
RESPONSE 12
- In 3.2 Variable Description; Author chooses the method used by Lusardi and Mitchell (2011a). The choice of three questions is beyond doubt. However, the choice of questions raises reservations. The author decided to apply questions regarding interest rates, discounting and borrowing. The article deals with investment choices of households, why did the author choose these questions and not others?
- Also, in the theoretical part, the author devotes a lot of space to the risk of an investment that is not addressed here. There is a visible lack of consistency between the title of the article and questions, as well as a description in the theoretical part and the empirical part.
RESPONSE 12
- The article deals with investment choices of households in relation to financial literacy. The three questions were asked in the FinScope (2017) survey, from which this paper adapts the definition of financial literacy and its data, to test whether participants have a general understanding of 3 basic concept of financial literacy namely; interest rates, discounting and borrowing.
- To avoid the inconsistency between the title of the article and questions, as well as a description in the theoretical part and the empirical part, all information related to risk of investment are ignored throughout the paper. The revised version of the attached paper does not have information related to risk of investment which brings the suggested inconsistency by the reviewer
COMMENT13
I consider the choice of the Binary probit regression method reasonable. The analysis is impressive and is made at a better level than the theoretical part. However, reading the article indicates some inaccuracies between the theoretical and empirical parts.
The theoretical part is quite general. It tackles investment issues, but not as much as the empirical part.
RESPONSE 13
A significant improvement is now done on the theoretical literature part. See a theoretical part from page 4-5
COMMENT 14
The article requires significant refinement and improvement. At the moment, it can be treated more like a research report than a scientific article.
RESPONSE 14
A significant refinement and improvement is now done in the attached revised version of the paper.
Round 2
Reviewer 1 Report
Dear authors,
I appreciate that you accepted my recommendations.
But please, follow IJFS template for your article in details as I mentioned in last review.
https://www.mdpi.com/journal/ijfs/instructions
Rename A Concluding Remark to Conclusions.
These articles could be added to enrich your paper:
Nguyen, T. A. N., Rozsa, Z. (2019). Financial Literacy and Financial Advice Seeking for Retirement Investment Choice. Journal of Competitiveness, 11(1), 70–83. https://doi.org/10.7441/joc.2019.01.05
Belás, J., Nguyen, A., Smrčka, L., Kolembus, J., Cipovová, E. (2016). Financial Literacy of Secondary School Students. Case Study from the Czech Republic and Slovakia, Economics and Sociology, Vol. 9, No 4, pp. 191-206. DOI: 10.14254/2071- 789X.2016/9-4/12
I hope, my comments may help you to improve the paper.
Author Response
COMMENTS
- I have changed section 5 title from A concluding remark to Conclusions- See page 12 of the attached revised version of the paper
- I appreciate for the proposed readings. They have enriched my paper; I. I have used the paper " Nguyen, T. A. N., Rozsa, Z. (2019). Financial Literacy and Financial Advice Seeking for Retirement Investment Choice.
Journal of Competitiveness,
- 11(1), 70–83. to improve my theoretical literature. See page 4 of the revised paper highlighted in blue and the reading listed in the reference and highlighted in blue color page 15
- ii. I have also used the paper " Belás, J., Nguyen, A., Smrčka, L., Kolembus, J., Cipovová, E. (2016). Financial Literacy of Secondary School Students. Case Study from the Czech Republic and Slovakia,
Economics and Sociology,
- Vol. 9, No 4, pp. 191-206" to improve my conclusion and highlighted the improvement in green color and also listed the reading in the references section highlighting it in green color, page 13
Reviewer 2 Report
The paper is much improved. Thank you for your efforts.
Author Response
I appreciate the professional review offered by the reviewer
Reviewer 3 Report
The article is much better after making changes to the text. I recommend to print.
Author Response
The professional review provided by the reviewer are highly appreciated